# Cryptococcal antigen carriage among HIV infected children aged 6 months to 15 years at Laquintinie Hospital in Douala

**Ginette Claude Mireille Kalla[1,2], Josette Farida Mboumnyemb[1], Jules Clément Nguedia Assob[3], Marcelle Nina Ehouzou Mandeng[1,2], Nelly Kamgaing Noubi[1,2], Marie Claire Okomo Assoumou[4], Francois-Xavier Mbopi-Keou**  **[4]\*, Francisca Monebenimp[1,2]**

**1** Department of Pediatrics, Faculty of Medicine and Biomedical Sciences, University of Yaoundé I, Yaoundé, Cameroon, **2** Yaoundé University Teaching Hospital, Yaoundé, Cameroon, **3** Department of Laboratory Medicine, Faculty of Medicine and Pharmaceutical Sciences, University of Douala, Douala, Cameroon, **4** Department of Microbiology, Parasitology, Heamatology and Infectious Diseases, Faculty of Medicine and Biomedical Sciences, University of Yaoundé I, Yaoundé, Cameroon

\* fxmkeou@hotmail.com

## Abstract

### Background

Up to 15% of deaths of people living with HIV is attributable to meningeal cryptococcosis, with nearly 75% occuring in sub-Saharan Africa. Although rare in children, it is a major cause of morbidity and mortality in people living with HIV. A strong association between cryptococcal antigenemia and the development of meningeal cryptococcosis has been shown in adults. Thus, in 2018, the World Health Organization published an updated version of its guidelines for the diagnosis, prevention and management of cryptococcal infection in adults, adolescents and the HIV-infected child.

### Goal

To determine the prevalence of cryptococcal antigenemia and to identify its determinants in children infected with HIV.

### Methods

An analytical cross-sectional study was carried out at the approved treatment center of Laquintinie hospital in Douala over a period of 4 months. Children were recruited consecutively after informed parental consent. Cryptococcal antigenemia and CD4 assay were performed using a Cryptops® immunochromatographic rapid diagnostic test and flow cytometry, respectively. The data collected included the socio-demographic, clinical and paraclinical variables of the children, as well as their antecedents. Data analysis was performed using Epiinfo software version 3.1 and SPSS 21.0. The significance threshold was set at 5%.

**Data Availability Statement:** All relevant data are within the manuscript.

**Funding:** The authors received no funding for this work.

**Competing interests:** The authors have declared that no competing interests exist.

## Results

A total of 147 children were enrolled. The mean age was 9.8 ± 4.09 years. The majority were on antiretroviral therapy (142, 96.60%). Only 13 (8.80%) were in severe immunosuppression. No child showed signs of meningeal cryptococcosis. The prevalence of cryptococcal antigenemia was 6.12%. Severe immunosuppression [OR: 10.03 (1.52–65.91), p = 0.016] and contact with pigeons [OR: 9.76 (1.14–83.65), p = 0.037] were independent factors significantly associated with the carriage of the cryptococcal antigen.

## Conclusion

We recommend screening for cryptococcal antigenemia and routine treatment with fluconazole of all HIV positive children with cryptococcal antigen whether symptomatic or not.

## Introduction

Cryptococcal meningitis is a serious opportunistic infection; it specifically occurs after *Cryptococcus* has spread from the lungs to the brain causing headache, fever, neck pain, nausea and vomiting, sensitivity to light, altered mental status (ranging from confusion to coma). Globally, it is estimated to be responsible for 15% of deaths of HIV-positive patients worldwide, with three quarter occurring in sub-Saharan Africa [1]. In 2014, cryptococcal meningitis, together with tuberculosis were by far the most common presentation associated with HIV/AIDS [2]. It was responsible for about 223,100 cases and 181,100 deaths among people living with HIV [2]. The literature reports that cryptococcal antigenemia is a strong predictor of subsequent cryptococcal meningitis in adults infected with Human Immunodeficiency Virus (HIV) with CD4 T cell counts <100/μl [3]. This is because cryptococcal antigen can be detected in the blood weeks or months before the development of clinical disease [1, 4, 5]. World Health Organisation guidelines for the diagnosis and prevention of HIV-associated cryptococcal meningitis recommends empirical treatment for any patient with persistent positive cryptococcal antigenemia, to avoid the development of a potentially fatal infection [6]. Screening for cryptococcal antigenemia coupled with preventive antifungal therapy has been shown to be a cost-effective strategy with survival benefits and has been incorporated into national HIV guidelines in several countries [6]. However, for some countries with a high HIV burden, this has not been implemented [6]. Many studies in Subsaharan Africa done among adults reported that the mortality rate of cases of cryptococcal meningitis was estimated at 35 to 65% in African patients infected with HIV, against 14 to 26% in HIV-infected patients living in industrialized countries [7, 8].

With the aim of reducing this morbidity and mortality associated with cryptococcal meningitis, in March 2018, WHO published an updated version of its guidelines for the diagnosis, prevention and management of cryptococcal infection in adults, adolescents and children infected with HIV [6]. Indeed, cryptococcal antigen screening is the preferred approach to identify the risk of cryptococcal meningitis development during the management of people with advanced HIV infection [2]. The implementation of the new WHO guidelines will help improve the diagnosis, prevention and treatment of one of the most common opportunistic infection in people with advanced HIV infection, thereby helping to reduce HIV-related mortality globally and, in particular, in Africa [9].

Many studies have looked at cryptococcal antigenemia, for instance, in 2017, Oladele et al. reported an average global prevalence of 6% cryptococcal antigenemia among HIV-infected patients with CD4 <100 cells/μl [1, 10].

In Cameroon, several studies have been carried out in adults, including that of Temfack et al. in 2018, which found a prevalence of cryptococcal antigenemia of 7.5%. In these patients with positive cryptococcal antigenemia, 45.5% of them developed meningeal cryptococcosis [10].

Although the literature is not very extensive, a few authors have focused on cryptococcosis in HIV positive children. All of these studies are unanimous in saying that meningeal crypto-coccosis is rare in children [2]. Indeed, the incidence of cryptococcal meningitis in children varies between 0.85 and 2.97% [11]. In Colombia, Lizarazo et al. found in 2014, an average annual incidence rate of the country of 0.017 cases/100,000 children under 16 years [12]. A study done in China by Guo et al. in 2016, found an incidence of cryptococcosis of 0.016 to 100 cases /100,000 children [13].

For this reason, WHO has limited this recommendation to adolescents and not to smaller children. According to WHO, screening and primary prophylaxis are not recommended for children, due to the low incidence of cryptococcal meningitis in this age group [13]. And yet, although the disease is not common in children, it remains a significant cause of morbidity and mortality, especially in those with weakened immune systems [10, 14, 15].

Thus, some authors have recommended screening for cryptococcal antigenemia in young children [12, 16].

In Cameroon, we did not find any studies on cryptococcosis in children, much less on the carriage of the cryptococcal antigenemia, hence our interest in the subject. Indeed, the early detection of the carriage of the cryptococcal antigenemia in Cameroonian HIV positive children, especially those with low CD4 count, would not only make it possible to assess the extent of cryptococcal antigenemia in the latter, but above all, to detect and treat early those likely to develop meningeal cryptococcosis which can be fatal.

## Materials and methods

A cross-sectional, analytical study was carried out in the pediatric department of Laquintinie hospital in Douala over a period of 4 months. The approved treatment center for people living with HIV in the said hospital has a cohort actively on treatment with more than 3,126 patients, including more than 800 children infected with HIV. The study population consisted of HIV positive children aged 6 months to 15 years of age whose parents had given written informed consent, they were enrolled consecutively.

### Data collection procedures

After having obtained the approval of the Ethics and Research Committee of the Faculty of Medicine and Biomedical Sciences of the University of Yaoundé I and the research authoriza-tion of the Director of the Laquintinie hospital in Douala, the children fulfilling our inclusion criteria were selected from the consultation registers. Parents were contacted by phone, invited to the hospital where information about the study was given to them followed by signature of an informed consent form for their child to participate in the study. Subsequently, all the chil-dren selected underwent a complete physical examination and the socio-demographic, clinical, paraclinical variables as well as the children's history were filled in using a pre-established questionnaire. At the end of this examination, a blood sample was taken in dry and EDTA tubes respectively for the determination of the cryptococcal antigen and that of CD4. The test-ing of cryptococcal antigenemia was done once a week for logistic reasons, however, for

children who were symptomatic, this was done immediately and all those who had a positive result were referred to a specialist consultation for management.

## Biological analysis procedures

After blood sample collection, tubes were stored in cooler boxes and transported to the laboratory within 24hours where they were centrifuged for 5 minutes, separated and stored at -20 degrees Celsius.

We used the semi-quantitative CryptoPS immunochromatographic test from Biosynex laboratories, Strasbourg, France. It is a rapid immunochromatographic test for the semi-quantitative detection and titration of Cryptococcus sp capsular antigen in serum, plasma, whole blood, and cerebrospinal fluid to guide the diagnosis of cryptococcosis [17]. With a sensitivity of 95% and a specificity of 100%. The T1 band is qualitative and the T2 band semi-quantitative [17]. The test was considered positive in the presence of two colored lines (T1 and control C) and strong positive in the presence of three colored lines (T1, T2 and C) (Fig 1).

During CrAg testing, the stored sera are brought to room temperature.

For the CD4 count, blood was taken in EDTA tubes and analysis was done by flow cytometry in a reference laboratory in the city of Douala. The classification of immunosuppression was made taking into account the number of $CD4/mm^3$ according to age groups as proposed by the WHO in 2006, classifying HIV immunosuppression in children into 4 classes (no significant, moderate, advanced and severe) [18].

## Data analysis

Data were entered into EPI data version 3.1 software and analyzed using Microsoft Excel 2016 and SPSS version 21.0 software. A simple calculation of proportions, means and standard deviation was performed. For comparison of proportions, chi-square and Fischer tests were used. The strength of association was estimated by Odds ratio and the 95% confidence interval. In order to exclude the effect of confounding factors, multivariable analysis was performed using the logistic regression model, including all variables with a p-value less than 0.05. The p value <0.05 was considered to be statistically significant.

## Ethical considerations

Confidentiality and anonymity were strictly observed. Parents did not pay any fees for the tests and were advised that their child may experience mild pain at the blood sample collection site. The samples were taken by qualified nursing staff, in accordance with the rules of asepsis. Positive cases of cryptococcosis were referred for treatment.

## Results

From January 22, 2018 to May 22, 2018, 147 children meeting our inclusion criteria were selected for the study. The mean age was 9 years 9 months ± 4.09 years with extremes ranging from 7 months to 15 years 8 months. The most represented age group was that of 10 to 15 years old with a frequency of 74 (50.30%). Seventy-four (74, 50.30%) were male for a sex ratio of 1.01. The majority of children lived in urban areas, ie 134/147 (91.2%). The majority had at least a primary school level (126, 85.7%) (Table 1).

## Peri and post-natal history of children

The majority 140 (95.20%) were born at term with a eutrophic birth weight (140, 70.70%). Slightly more than half benefited from exclusive breastfeeding (78, 53.10%) and 72 (49%) of

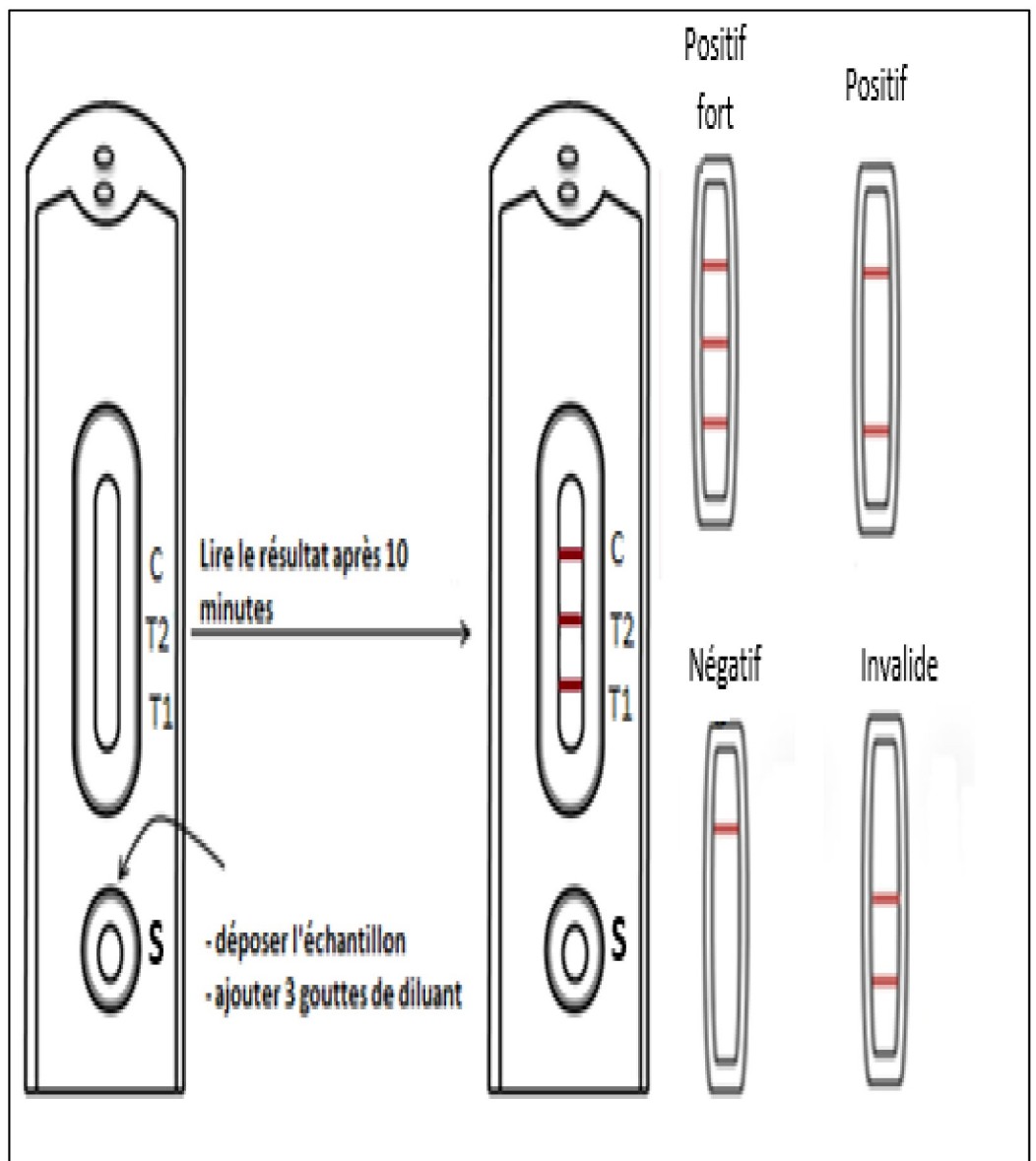

**Fig 1. Interpretation of the results of Cryptops®test.** Cryptococcal antigen carriage among children aged 6 months to 15 years infected with HIV at Laquintinie Hospital in Douala.

them were weaned before the age of 6 months. HIV status was unknown for 100 (68%) mothers during pregnancy and the most common mode of delivery was vaginally (145/147, 98.60%). The majority, 99 of the children had a history of previous hospitalization (67.40%) and 118 (80.30%) of them had no history of opportunistic illnesses. For those who did, pulmonary tuberculosis was the most common opportunistic disease (22/147, 15%). Most of the children were not in contact with domestic birds (129, 87.80%). However, 11 children were in contact with chickens (7.50%), 5 children with pigeons (3.40%) and finally, 2 children with ducks (1.40%) (Table 2).

**Table 1. Sociodemographic characteristics of children infected with HIV at Laquintinie hospital in Douala.**

| Variables | Frequency (n) | Percentage (%) |
|---|---|---|
| **Age (years)** | | |
| < 1 | 2 | 1.40 |
| [1–3] | 9 | 6.10 |
| [3–5] | 11 | 7.50 |
| [5–10] | 51 | 34.70 |
| [10–16] | 74 | 50.30 |
| **Sex** | | |
| Male | 74 | 50.30 |
| Female | 73 | 49.70 |
| **Place of residence** | | |
| Urban | 134 | 91.20 |
| Rural | 13 | 8.80 |
| **Level of education** | | |
| Unschooled | 6 | 4.10 |
| Kindergarten | 15 | 10.20 |
| Primary | 77 | 52.40 |
| Secondary | 49 | 33.30 |
| **Total** | **147** | **100** |

## Clinical and paraclinical characteristics of pediatric HIV

Seventy-nine children had been known to be HIV positive for more than 59 months (53.70%) and 74 (50.30%) had been on ARVs for more than 5 years. Only 22 (15%) benefited from a change in treatment protocol following treatment failure for 9 (6.10%) of them. An adherence problem was identified in 73 children (49.70%), but the majority of children did not interrupt their treatment 143 (97.30%).

The majority of children at the time of the study were asymptomatic (138, 94.60%). However, we found 3 children with otorrhea, 2 children with signs of malnutrition, 2 children with fever, 1 with headache and 1 other who presented with vomiting. The viral load was undetectable for 84 children (57.10%) and the immune deficiency was moderate, advanced and severe for 24 (16.30%), 17 (11.60%) and 13 (8.80%) respectively (Table 3).

## Cryptococcal antigen carriage

Of the 147 children selected for the study, 9 children were carriers of the cryptococcal antigen, giving a prevalence of 6.12%. The antigenic titers were very high (> 2.5 µg / ml) for 2 (22.22%) children and between moderate (2.5 ng-2.5 µg/ml) for 7 (77.78%) children (Table 4).

## Factors associated with the carriage of cryptococcal antigen

Discovery of HIV status less than 2 months [OR = 9.57; IC95% (1.03–61.96); (p = 0.040)], discontinuation of ARVs [OR = **19.43, IC 95% (1.7–198.07)** (p = 0.020)], ARVs taking less than 2 months [OR = **21.28, IC95% (3.79–107.34)** (p = 0.000)], regular contact with pigeons [OR = 12.86, IC 95% (1.29–95.96) (p = 0.030)], the presence of clinical manifestations [OR = 36, IC 95% (5.55–223.64) (p = 0.000)] and severe immunosuppression [OR = **12.86, IC 95% (1.29–95.96)** (p = 0.000)] were significantly associated with the carriage of the cryptococcal antigen.

**Table 2. Perinatal and postnatal history of mothers and children.**

| Variables | Fréquence (n) | Pourcentage (%) |
|---|---|---|
| **Gestational age (WA)** | | |
| <37 | 6 | 4.10 |
| [37–42] | 140 | 95.20 |
| >42 | 1 | 0.70 |
| **Weight at Birth (Gr)** | | |
| Not Known | 37 | 25.20 |
| <2500 | 3 | 2.00 |
| [2500–4000] | 104 | 70.70 |
| >4000 | 3 | 2.00 |
| **Feeding method at birth** | | |
| Not known | 12 | 8.20 |
| Exclusive breastfeeding | 78 | 53.10 |
| Artificial feeding | 18 | 12.20 |
| Mixed feed | 39 | 26.50 |
| **Age at weaning (months)** | | |
| Not known | 12 | 8.20 |
| <6 | 72 | 49.00 |
| [6–9] | 33 | 22.40 |
| >9 | 30 | 20.40 |
| **HIV status of the mother during pregnancy** | | |
| Not known | 100 | 68.00 |
| Positive | 13 | 8.80 |
| Negative | 34 | 23.10 |
| **Mode of delivery** | | |
| Per vaginal | 145 | 98.60 |
| Caesarean | 2 | 1.40 |
| **Total** | 147 | **100** |
| **Postanatal** | | |
| **Variables** | **Frequency (n)** | **Percentage (%)** |
| **Previous hospitalizations** | | |
| Any | 48 | 32.60 |
| ≥ 1 | 99 | 67.40 |
| **History of opportunistic diseases** | | |
| No history | 118 | 80.30 |
| Pulmonary tuberculosis | 22 | 15.00 |
| Kaposi sarcoma | 1 | 0.70 |
| Shingles | 1 | 0.70 |
| Toxoplasmosis | 1 | 0.70 |
| Oropharyngeal candidiasis | 4 | 2.70 |
| **Contact with pets** | | |
| No contact | 129 | 87.80 |
| Chickens | 11 | 7.50 |
| Pigeons | 5 | 3.40 |
| Ducks | 2 | 1.40 |
| **Total** | **147** | **100** |

WA: weeks of amenorrhea, Gr: Grams.

**Table 3. Clinical and paraclinical characteristics of pediatric HIV.**

| Variables | Frequency (n) | Percentage (%) |
|---|---|---|
| **Duration of HIV discovery (months)** | | |
| <2 | 6 | 4.10 |
| [2–3] | 7 | 4.80 |
| [4–12] | 16 | 10.90 |
| [13–59] | 39 | 26.50 |
| > 59 | 79 | 53.70 |
| **Duration of ARV intake (months)** | | |
| Naive | 5 | 3.40 |
| <2 | 5 | 3.40 |
| [2–59] | 63 | 42.90 |
| > 59 | 74 | 50.30 |
| **Changing ARV treatment** | | |
| Yes | 22 | 15.00 |
| No | 125 | 75.00 |
| **Therapeutic failure** | | |
| Yes | 9 | 6.10 |
| No | 138 | 93.90 |
| **Compliance issues** | | |
| Yes | 73 | 49.70 |
| No | 74 | 50.30 |
| **Discontinuation of treatment** | | |
| Yes | 4 | 2.70 |
| No | 143 | 97.30 |
| **Clinical manifestations** | | |
| Yes | 9 | 5.40 |
| No | 138 | 94.60 |
| **Viral load (copies / ml)** | | |
| Not carried out | 20 | 12.90 |
| Undetectable (less than 50) | 84 | 57.10 |
| [51–199] | 5 | 3.40 |
| [200–299] | 2 | 1.40 |
| [300–500] | 2 | 1.40 |
| > 500 | 35 | 23.80 |
| **Immune deficiency** | | |
| Not significant | 93 | 63.30 |
| Moderate | 24 | 16.30 |
| Advanced | 17 | 11.60 |
| Strict | 13 | 8.80 |
| **Total** | **147** | **100** |

ARVs: antiretrovirals; ml: milliliter.

## Multivariable analysis

Regular contact with pigeons [OR = 9.76, IC95% (1.14–83.65) (p = 0.037)] and severe immunosuppression (OR: 10.03 [1.52–65.91], p = 0.016) [OR = 10.03, IC95% (1.52–65.91) (p = 0.016)] were found to be independent factors significantly associated with the carriage of the cryptococcal antigen in HIV positive children (Table 5).

**Table 4. Distribution of the population according to prevalence of cryptococcal antigenemia and to antigenic titre.**

| Variables | Frequency (n) | Percentage (%) |
|---|---|---|
| **Presence of cryptococcal** | | |
| antigen in serum | | |
| Positive | 9 | 6.12 |
| Negative | 138 | 93.88 |
| **Total** | **147** | **100** |
| Antigenic titre in serum | | |
| [25ng-2.5µg] | 7 | 77.78 |
| > 2.5µg | 2 | 22.22 |
| **Total** | **9** | **100** |

## Discussion

Most participants had history of previous hospitalization (67.40%), although118 (80.30%) presented no history of opportunistic illnesses (118 (80.30%) in this series. For those who did, pulmonary tuberculosis was the most common opportunistic disease (22/147, 15%). However, 11 children were in contact with chickens (7.50%), 5 children with pigeons (3.40%) and, 2 children with ducks (1.40%). Liu *et al.* in his pediatric cohort population study carried out in China in 2017, found that 19 children (35.8%) had history of exposure to poultry including 11 children exposed to pigeons and 8 exposed to chickens [14]. Numerous other studies have suggested that *Cryptococcus neoformans* is found worldwide in association with the feces of certain birds such as pigeons [19, 20].

The majority of children in our series were on ARVs (142, 96.60%). This was also the case in the study by Somdipa *et al.* in India [5]. Most children at the time of the study were

**Table 5. Multivariable analysis of factors associated with carriage of cryptococcal antigen.**

| Variables | Cryptococcal antigen | | OR (IC 95%) | P value |
|---|---|---|---|---|
| | Positive (n = 9) | Negative (n = 138) | | |
| **Duration of HIV discovery <2 months** | | | | |
| Yes | **2 (33.3)** | **4 (66.7)** | **9.57(1.03–61.96)** | **0.040** |
| No | 7 (5) | 134 (95) | | |
| **Duration of ARV intake <2 months** | | | | |
| Yes | **4 (44.4)** | **5 (55.6)** | **21.28(3.79–107.34)** | **0.000** |
| No | 5 (3.6) | 133 (96.4) | | |
| **Not taking ARVs** | | | | |
| Yes | **2 (50)** | **2 (50)** | **19.43(1.7–198.07)** | **0.020** |
| No | 7 (4.9) | 136 (95.1) | | |
| **Regular contact with pigeons** | | | | |
| Yes | **2 (40)** | **3 (60)** | **9.7(1.29–95.96)** | **0.037** |
| No | 7 (4.9) | 135 (95.1) | | |
| **Clinical manifestations** | | | | |
| Yes | 4 (57.1) | **3 (42.9)** | **36(5.55–223.64)** | **0.000** |
| No | 5 (3.6) | 135 (96.4) | | |
| **Immunodepression strict** | | | | |
| Yes | **4 (30.8)** | **9 (69.2)** | **10.03(2.29–51.64)** | **0.016** |
| No | 5 (3.7) | 129 (96.3) | | |

asymptomatic (138, 94.60%). However, few children presented signs of otorrhea (03), signs of malnutrition (02), with fever (02), headache (01) and vomiting (01). In the literature, although reported in adults, headache seems to be a predictor of cryptococcal meningitis in patients with cryptococcal antigen (p <0.001). Regarding malnutrition, Goni *et al.*, in 2017 in Nigeria in a study carried out in adults reported that low body mass index was an independent predictor of positive serum cryptococcal antigenemia (p = 0.037) [21].

We found a prevalence of cryptococcal antigenemia of 6.12%. Anigilaje *et al.* in 2013 in Nigeria and Somdipa *et al.* in 2019 in India found that no child was carrying a cryptococcal antigen [5, 22]. The prevalence found in our series is much closer to that found in studies carried out in HIV-positive adults, in particular Oladele *et al.* in Nigeria in 2016 and Temfack et al. in 2018 in Cameroon who found a prevalence of cryptococcal antigenemia respectively of 8.9% and 7.5% [7, 9]. If the results of Anigilaje *et al.* [22], as well as Somdipa *et al.* [5] are in line with the WHO recommendations excluding adolescents and children from routine screening for cryptococcal antigenemia, our results on the other hand suggests that systematic screening is recommended for children.

Significant factors associated with the carriage of the cryptococcal antigen such as discovery of HIV status less than 2 months (p = 0.040), discontinuation of ARVs (p = 0.020), ARVs taking less than 2 months (p = 0.000), regular contact with pigeons (p = 0.030), the presence of clinical manifestations (p = 0.000) and severe immunosuppression (p = 0.000) were revealed in this study. However other similar studies in children did not find any factors associated with the carriage of the cryptococcal antigen [5, 22].

However, many studies carried out in adults, in particular the study by Ogouyèmi-Hounto et al. in Benin in 2016, found as a factor associated with the carriage of the cryptococcal antigen, the body mass index <18.5 kg/m$^2$ and an altered general condition with a CD4 lymphocyte count <50 cells / µL [23]. Hailu *et al.* in Ethiopia in 2019 found that male sex, rural life and being hospitalized were associated with cryptococcal antigenemia [24].

## Conclusion

The prevalence of carriage of the cryptococcal antigen was 6.12%. Severe immunosuppression and contact with pigeons were independent factors significantly associated with this carriage. At the end of this study, we recommend screening for cryptococcal antigenemia and routine treatment of all HIV positive children with cryptococcal antigen, whether symptomatic or not. Currently, no study on neuromeningeal cryptococcosis and the carriage of the cryptococcal antigen in children in Cameroon. Very little data on the carriage of the cryptococcal antigen in children worldwide.

This study shows that the carriage of the cryptococcal antigen is a reality in HIV positive children in Cameroon. Screening for cryptococcal antigenemia and preemptive treatment with fluconazole should also be routine in advanced HIV positive children, contrary to the 2018 WHO recommendations on the prevention, diagnosis and management of cryptococcosis in HIV positive subjects.

## Supporting information

**S1 Dataset.**
(XLSM)

**S2 Dataset.**
(XLSM)

## Acknowledgments

Our thanks go to all the patients and parents for their participation in the study, as well as to Prof. Penda Ida and the staff of Laquintinie Hospital in Douala.

## Author Contributions

**Conceptualization:** Ginette Claude Mireille Kalla.

**Formal analysis:** Ginette Claude Mireille Kalla, Josette Farida Mboumnyemb, Jules Clément Nguedia Assob, Marcelle Nina Ehouzou Mandeng, Nelly Kamgaing Noubi, Marie Claire Okomo Assoumou.

**Investigation:** Josette Farida Mboumnyemb.

**Project administration:** Ginette Claude Mireille Kalla.

**Resources:** Francisca Monebenimp.

**Supervision:** Francisca Monebenimp.

**Validation:** Jules Clément Nguedia Assob.

**Writing – original draft:** Josette Farida Mboumnyemb, Jules Clément Nguedia Assob, Marcelle Nina Ehouzou Mandeng, Nelly Kamgaing Noubi, Marie Claire Okomo Assoumou.

**Writing – review & editing:** Francois-Xavier Mbopi-Keou, Francisca Monebenimp.

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
