## [Decision Letter · Decision Letter 0]

25 Feb 2021

PONE-D-20-40134

Cryptococcal antigen carriage among HIV infected children aged 6 months to 15 years at Laquintinie Hospital in Douala

PLOS ONE

Dear Dr. Mbopi-Keou,

Thank you for submitting your manuscript to PLOS ONE. After careful consideration, we feel that it has merit but does not fully meet PLOS ONE’s publication criteria as it currently stands. Therefore, we invite you to submit a revised version of the manuscript that addresses the points raised during the review process.

1. Required: Respond point-by-point to the reviewers' comments

2. Recommended: Edit manuscript for proper English language usage by having a native english speaker read and edit the manuscript.

We look forward to receiving your revised manuscript.

Kind regards,

Kirsten Nielsen, Ph.D

Academic Editor

PLOS ONE

Journal Requirements:

Reviewers' comments:

Reviewer's Responses to Questions

**Comments to the Author**

1. Is the manuscript technically sound, and do the data support the conclusions?

Reviewer #1: Partly

2. Has the statistical analysis been performed appropriately and rigorously? 

Reviewer #1: No

3. Have the authors made all data underlying the findings in their manuscript fully available?

Reviewer #1: No

4. Is the manuscript presented in an intelligible fashion and written in standard English?

Reviewer #1: No

5. Review Comments to the Author

Reviewer #1: Dear authors,

Thank you for the opportunity to review the manuscript entitled "Cryptococcal antigen carriage among HIV infected children aged 6 months to 15 years at Laquintinie Hospital, Douala" which I read with much attention. The authors opted to address the prevalence of cryptococcal antigenemia (CrAg) in children aged 6 months to 15 years in an HIV treatment centre in a reference hospital in Douala, the economic capital of Cameroon. They found a prevalence of 6.12%, similar to what has been found in adults with HIV and severe immune depression. With these findings, the authors go further to recommend systematic CrAg screening in children with severe immune depression associated with HIV. The manuscript is of interest addressing a strategy to manage an opportunistic infections to reduce HIV-associated mortality. However, I have made some comments with the aim of contributing to making the manuscript better for the readership of the journal. My comments include:

Introduction

- Line 73: The first sentence stating that "Cryptococcosis meningitis is a serious opportunistic infection" seems barren because such a statement needs back up statements and references to explain why and how serious the infection is. Please kindly review.

- Line 73: The first two words should be "Cryptococcal meningitis" NOT "Cryptococcosis meningitis".

- Line 74: please write 3/4 in full (three quarters) or express in percentage

- Line 75 to 76: Saying cryptococcal meningitis "was by far the most common presentation associated with HIV/AIDS" is not true because according to UNAIDS data, cryptococcal meningitis is the second common opportunistic infection in HIV/AIDS after tuberculosis. Please kindly review.

- Line 82: Please review the statement "Many authors already prescribed empirical treatment". Authors do not prescribe, a better statement would be that "World health organisation guidelines for the diagnosis and prevention of HIV-associated cryptococcal meningitis recommend…….."

- Lines 84 to 86: This statements needs references because stated this way, it sounds as if it was said by the authors of this manuscript, which is not the case

- Lines 86 to 87: The statement saying cryptococcal antigen (CrAg) screening has not been implemented in some countries with high HIV burden, also needs a reference.

- Lines 87 to 91: Beginning the statement with "Thus" gives the impression that the study by Oladele was carried out because CrAg screening was not implemented in HIV high burden countries, which could be the case but in the statement, the authors mentioned mortality instead of CrAg prevalence. Please kindly review the word "thus".

- Line 87 to 91: Using Oladele et al as the only study to mention cryptococcal meningitis mortality in Africa as compared to industrialised countries is not enough because there have been many cohort in Africa expressing mortality related to cryptococcal meningitis even in the Cameroon context, that are worth citing here to make the argument stronger. Please check more references.

- Line 92 to 95: please add a reference to this updated WHO guidelines.

- Lines 95 to 97: What do the authors mean by "the risk of disease progression". I guess they meant the risk of cryptococcal meningitis, the most fatal presentation of cryptococcosis.

- Lines 102 to 104: The authors say many studies have looked at cryptococcal antigenemia but go on to cite only one study. Please cite a few more studies

- Line 125 to 126: I will suggest the authors to replace the word "carriage of cryptococcal antigen" with " cryptococcal antigenemia" because I suppose carriage here means in blood.

Materials and methods

- Lines 133 to 134: Can the authors kindly explain what "active file" means? I guess it was borrowed from French, in which case it could be replaced with the word "cohort actively on treatment"

- Lines 135 to 137: How was the informed consent done? Was it verbal or written? Please kindly specify.

- Lines 137 to 138: Why did the authors make the precision that recruitment was "consecutive and not probabilistic", one is not the opposite of the other, please kindly review.

- Line 142 to 143: Please review the statement "..the children responding to our inclusion criteria", replacing the word "responding" with "fulfilling" sounds better.

- Lines 142 to 146: The authors report that the parents of the children were contacted by phone which may imply that parents without phone numbers were excluded, an approach which in my opinion creates selection bias. Secondly, the statement presented the way it is, gives the impression that information was given by phone. I will suggest that the authors review the statement to mention clearly that the parents were contacted by phone, invited to the hospital where information about the study was given to them followed by signature of an informed consent form for their child to participate in the study.

- Line 151 to 154: I will suggest replacing the word "dosage" with "testing".

- Line 151 to 154: Why did the authors decide to do testing once a week with a point of care test? Was it for logistic reason? If that was the case it should be clearly stated.

- Line 156 to 156: the sentence has repeated words that make understanding difficult. I will suggest rephrasing to "After blood sample collection, tubes were stored in cooler boxes and transported to the laboratory within 24hours where they were centrifuged for 5 minutes, separated and stored at -20 degrees Celsius".

- Lines 158 to 159: A repetition of the fact that the test were done once a week. Please kindly explain why with a point of care test, it was chosen to be done once a week.

- Lines 161 to 164: "…to guide the diagnosis of cryptococcus, especially meningitis". I will suggest reviewing this statement by changing cryptococcus to cryptococcosis and deleting "especially meningitis".

- Lines 164 to 166: The threshold of detection for T1 and T2 bands in my opinion is not relevant stating here because these details would unintentionally serve here to confuse the uninformed reader. I think that simply stating T1 as qualitative and T2 as semi-quantitative (appearing only when there is high antigen titre) is enough as rightly stated in lines 166 to 167.

- lines 170 171: I guess this is the testing procedure. Statement as currently in the manuscript is confusing. I will suggest to say, "During CrAg testing, the stored sera are brought to room temperature……"

- Lines 171 to 172: "The blood collected in the dry tubes was centrifuged for 5 minutes to obtain serum" is a repetition of what was already stated above.

- Lines 171 to 174: Describing the testing procedure in my opinion is irrelevant here because in the previous statement in lines 170 to 171, the authors already mentioned that the tests were performed according to manufacturer's procedure.

Data analysis

- Lines 184 to 185: please kindly delete the word "the" before the word "odd ratio"

- Line 185 to 187: Please kindly replace "multivariate" with "multivariable"

- Lines 185 to 187: During the multivariable analysis why did you choose a threshold p-value of 0.05 for including variables in the model after the univariable analysis. Conventionally, people use 0.1. Please kindly explain.

- Lines 190 to 195: The content of this section is a repetition of what was already mentioned in lines 140 to 146, please kindly review.

- Lines 195 to 196: I will suggest to replace "test site" with "blood sample collection site" or "phlebotomy site"

- Lines 197 to 198: review the word "cryptococcuses"

Results

- Table 2: Mode of delivery, please review the word "low way", I guess you mean "per-vaginal delivery"

- Lines 244 to 245: what was the threshold of HIV viral load detectability.

- Lines 244 to 247: Though CD4 is always expressed for age, it could be interesting for the authors to state the median CD4 for the study population to permit the reader create a personal impression on the immune status of the study population because stating it the way it is currently written is vague.

- Line 251: I will suggest replacing "carrying of the cryptococcal antigen" with "prevalence of serum cryptococcal antigen". In this section, it will be interesting to state some relevant characteristics of those who were CrAg positive in the study.

- Lines 260 to 265: Please kindly state in the text the odd ratios and 95% confidence intervals, reporting only the p-values is unconventional and makes no sense to the reader

- Line 264 to 265: what do the authors mean by "..we found a trend towards significance for sex?". Please kindly explain further because the only information shown in the text is a p-value of 0.06.

- Table 6 and 7: the odd ratios are very large and their 95% Confidence intervals are very wide with the lower value around 1, could the authors comment on this? Do they think it is a true evaluation of the presence of association or the low p-values are due to chance associated with small numbers in their findings?

Discussion

- Lines 277 to 283: The authors cited studies in other settings where the male sex was predominant as compared to their study. Firstly, this raises a concern on whether a sex ratio of 1.01 expresses any predominance of one sex on the other. Secondly, it is not expected to discuss the sex ratio of the study population, I rather expected a discussion of the sex ratio among those who were found CrAg positive, which is the main point of interest of the study. Please kindly review. Thirdly, saying that predominance of males in Somdipa's study in India is due to the fact that girls are neglected, seems too harsh to me and I will suggest using softer language.

- Lines 284 to 286 is a repetition using the exact same words as in the results found on lines 223 to 225 which creates redundance in the manuscript, please kindly review this.

- Lines 287 to 289 is also a repetition of what is reported in lines 226 to 228 and requires to be reviewed.

- Lines 287 to 294: The authors discuss the contact with pigeons in their study with respect to the study by Liu et al in China and conclude that numerous studies have shown an association between contact with pigeons and cryptococcosis which is known but has not been shown in this study because looking at table 6, there were five children with a history of contact with pigeons, among whom 2 were CrAg positive and 3 were CrAg negative. More so, the authors reported that 9 children were found CrAg positive but in table 6, reporting the variable on contact with pigeons, the total number is 11 and total study population to 150 instead of 147 as reported, giving a computed odd ratio of 10.1 instead of the reported 12.86. Can the authors review?

- Lines 301 to 306: The two children who showed signs of malnutrition, what were their CrAg status? I really do not understand why the authors choose to discuss malnutrition as a factor related to CrAg positivity when they did not explore this in the study.

- Lines 309 to 312: The association between HIV-associated severe immune depression and cryptococcosis was established very early on at the start of the HIV pandemic which brings us to question the studies mentioned to not have found an association

- Line 328 to 332: This is a copy paste of the results, please review.

- Line 335 to 340: please kindly refrain from discussing factors that were not explored in your study.

- Line 343 to 346: It is also a copy paste of the results.

General comments:

- Tables should be labelled using Arabic not roman numerals.

- There are many tables (7), most of which have three columns showing only numbers and percentages with so much information many of which may not be relevant to the research question being treated. The authors should review the number of tables, I will suggest to bring to a maximum of four tables as well as reviewing the content, to express only information relevant to the topic. Moreover most of the tables are either too long or too short.

- In the tables, the way the intervals are presented is non-conventional with open ended square brackets, please kindly review.

- The results section has been divided into many subheadings which could have been combined under one main subheading, please kindly review.

- The HIV history of the study participants is very difficult to follow. The authors report that 68% of the mothers did not know their HIV status during pregnancy in a study population where 91% live in an urban area. This gives the impression of so many gaps in the PMTCT program in this region, most especially as 98% of these women delivered per-vaginally. Can the authors kindly explain at what time with respect to the pregnancy was the HIV status of the mothers ascertained? Was it done retrospectively after the child was confirmed HIV positive? Moreover, it is not clear under what circumstances the HIV status of the children was ascertained, this leaves the reader thinking that it was probably secondary to clinical signs of opportunistic infections though the authors say in line 224 that 80% of the children were asymptomatic, which somehow is contrasting with line 223 where they say 67% of the children had a history of hospitalisation. Going further to line 237, the authors report that 53% of the children were known HIV positive for 59months and 50% had been on ARV for more than five years, which gives the impression that most of the children were diagnosed at early childhood considering the mean age of the study population, which still goes further to iterate the need for a clear explanation on the circumstances of HIV status ascertainment in these children.

- In the discussion section, it is expected that the authors discuss findings in those found to be CrAg positive, instead, they discuss more of the characteristics of the study population

- In the discussion, many sections of the results are copied and pasted word for word as written in the results section.

6. PLOS authors have the option to publish the peer review history of their article (what does this mean?). If published, this will include your full peer review and any attached files.

Reviewer #1: **Yes: **Elvis Temfack

---

## [Author Response · Author response to Decision Letter 0]

6 Apr 2021

UNIVERSITE DE YAOUNDE I 

The University of Yaounde I 

FACULTE DE MEDECINE ET DES SCIENCES BIOMEDICALES 

Faculty of Medicine and Biomedical Sciences

DEPARTEMENT DE MICROBIOLOGIE-PARASITOLOGIE-HEMATOLOGIE-IMMUNOLOGIE-MALADIES INFECTIEUSES

Department of Laboratory Medicine, Microbiology, Hematology, Immunology and Infectious Diseases

Yaounde, April 2, 2020

Le Chef de Département

The Department Chair

To the Editor in Chief PlosOne

RE: Revised MS: Reference D-20-40134

Dear Sir:

We are pleased to resubmit our manuscript entitled: ‘Cryptococcal antigen carriage among HIV infected children aged 6 months to 15 years at Laquintinie Hospital in Douala Short title: Cryptococcal infection among HIV infected children in Cameroon’ for your appraisal after effecting all the corrections as suggested by the Reviewers. 

The corrections have been effected using track change and a clean version is provided as well.

A point by point correction done in order to answer the Reviewers requests is attached.

Yours Sincerely

Prof. François-Xavier MBOPI-KEOU, 

DBiol, MPhil, MSc, DLSHTM, Ph.D (London), MDP (Harvard), 

University Professor/Professeur Titulaire des Universités

P.O. Box: 3601 Yaounde-Cameroon; Tel: +237 699 95 80 15

Email: fxmkeou@hotmail.com

---

## [Editor Report · Decision Letter 1]

14 Jun 2021

Cryptococcal antigen carriage among HIV infected children aged 6 months to 15 years at Laquintinie Hospital in Douala

PONE-D-20-40134R1

Dear Dr. Mbopi-Keou,

We’re pleased to inform you that your manuscript has been judged scientifically suitable for publication and will be formally accepted for publication once it meets all outstanding technical requirements.

Kind regards,

Kirsten Nielsen, Ph.D

Academic Editor

PLOS ONE

Additional Editor Comments (optional):

I am sorry for the delay. I was waiting for a reviewer response.
---

## [Editor Report · Acceptance letter]

29 Jun 2021

PONE-D-20-40134R1 

Cryptococcal antigen carriage among HIV infected children aged 6 months to 15 years at Laquintinie Hospital in Douala 

Dear Dr. Mbopi-Keou:

I'm pleased to inform you that your manuscript has been deemed suitable for publication in PLOS ONE. Congratulations! Your manuscript is now with our production department. 

Kind regards, 

on behalf of

Dr. Kirsten Nielsen 

Academic Editor

PLOS ONE